# Effect of Shot Peen Forming on Corrosion-Resistant of 2024 Aluminum Alloy in Salt Spray Environment

**DOI:** 10.3390/ma15238583

**Published:** 2022-12-01

**Authors:** Jingzhen Qiao, Xiaowen Zhang, Guoqing Chen, Wenlong Zhou, Xuesong Fu, Junwei Wang

**Affiliations:** 1School of Materials Science and Engineering, Dalian University of Technology, Dalian 116024, China; 2Dalian Yuchen High Tech Material Co., Ltd., Dalian 116023, China

**Keywords:** shot peen forming, salt spray corrosion, intergranular corrosion, electrochemical test

## Abstract

The effect of shot peen forming on the corrosion-resistant of 2024 aluminum alloy in a salt spray environment was studied with an electrochemical workstation. The surface morphology and cross sectional morphology of the original and shot peen-formed sample were studied by a scanning electron microscope. After shot peen forming, the salt spray corrosion resistance of 2024 aluminum alloy was worsened (the corrosion rates of the original alloy and the shot peen-formed alloy were 0.10467 mg/(cm^2^·h) and 0.27333 mg/(cm^2^·h), respectively, when the salt spray corrosion time was 5 h). The radius of capacitive reactance arc of the sample subjected to shot peen forming was smaller than that of the original sample. When the salt spray corrosion time was 5 h, the doping density (*N*_A_) of the original alloy was 2.5128 × 10^−13^/cm^3^. After shot peen forming, the *N*_A_ of the alloy increased to 15 × 10^−13^/cm^3^. For the shot peen-formed sample, pitting corrosion first occurred in the crater lap zone and became severe with salt spray time. The cross sectional morphology of both original and the shot peen-formed samples shows that severe intergranular corrosion occurred in the salt spray environment. However, for the original sample, the intergranular corrosion distribution was lamellar. For shot peen-formed sample, the intergranular corrosion distribution was network.

## 1. Introduction

Aluminum alloy has low specific gravity, high specific strength, good electrical conductivity, strong corrosion resistance, good mechanical properties and low cost. Hence, it is widely used in aerospace, transportation, sports and shipping [1,2,3]. Among aluminum alloys, 2024 aluminum alloy, an Al-Cu-Mg aluminum alloy, is a kind of deformed aluminum alloy. Due to its high specific strength, excellent fatigue resistance and good formability, it is often used as the material for aircraft wall panel parts, such as the skin, frame, wing rib and so on [4,5]. In terms of the practical application of 2024 aluminum alloy wall panels, shot peening is constantly adopted to improve their corrosion resistance. Shot peening has a short production cycle, small site occupation area and strong adaptability to part size without a forming die. Therefore, it has become a preferred technical method for manufacturing modern large, light and high-strength aluminum alloy integral wall panels [6]. Moreover, shot peening is a kind technology of plastic deformation which introduces impurities induced from shot media, e.g., the introduction of Fe from stainless steel balls and iron pins [7,8]. Shot peen forming is used for the integral wall panel forming of the large wings of the Boeing 747 and Airbus A380 aircraft [9].

Shot peening includes shot peen forming and shot peen strengthening. The difference between shot peen forming and shot peen strengthening is that the former uses a larger shot medium. Consequently, compared with shot peen strengthening, the strength of shot peen forming is higher, and the degree of local strain on a surface is also higher with surface plastic deformation.

Surface plastic deformation also has an influence on the corrosion resistance. Low energy laser shock peening can enhance the corrosion resistance of aluminum alloy 7075 T651 and degrade the formation of microfracture and crack growth [10]. Researchers have accredited this to grain refinement, compressive residual stress, micro strain and dislocation density. The group of Wang [11] also believed that low shock peening results in the prevention of the formation and growth of corrosion pit. This is mainly due to grain refinement. During the corrosion stage, there will be more grains to bear the crack-driving force so that the corrosion rate decreases. Sun et al. [12] found that after ultrasonic shot peening, the exfoliation susceptibility of AA 7150 Al alloy decreased and intergranular corrosion was inhibited for both normal and transverse planes, pitting potential shifts to the positive direction.

In addition, the surface plastic deformation of the shot-peened sample will introduce compressive residual stress, which can effectively offset the external thrust generated by grain-boundary corrosion products and reduce the corrosion depth [13]. In addition, the compressive residual stress on surface layer can decrease the depth of stress corrosion cracking [14]. Moreover, shot peen forming can refine grains, and then increase the grain boundary area [15]. Structural refinement provides diffusion channels to form a passive film, increasing corrosion resistance [16]. However, Wu et al. [16] also believed that shot peening also increases the surface roughness of samples, which has negative effects on the corrosion resistance. Meanwhile, plastic deformation also introduces adverse factors, such as contamination [17,18] and surface microcrack [19]. As a result, the influence mechanism of surface plastic deformation on the corrosion resistance of aluminum alloy is complex.

In this paper, the effect of shot peen forming on the corrosion resistance of aluminum alloy sheets in a salt spray environment was studied regarding the surface morphology and the cross-section morphology. Combined with electrochemical impedance spectroscopy and Mott–Schottky results, the salt spray corrosion process of 2024 aluminum alloy formed by shot peen forming was analyzed.

## 2. Experimental Procedure

### 2.1. Materials and Chemicals

The sample material was 2024 aluminum alloy, which is a high-strength hard aluminum alloy, whose heat treatment process was natural aging after solid solution treatment. The chemical composition of the 2024 aluminum alloy was measured by an X-ray fluorescence spectrometer (Panaco, The Netherlands), as shown in Table 1.

The surface treatment process of the sample was shot peen forming and strengthening. The MP20000 numerical controlled shot peening machine (Shanghai Kaixin Machinery Manufacturing Co., Ltd., Shanghai, China) was utilized for shot peen forming and strengthening of the aluminum sample with cast steel shot pellets. The process parameters of shot peen forming and strengthening are shown in Table 2.

### 2.2. Methods

#### 2.2.1. Salt Spray Corrosion Test

Before the salt spray corrosion test, the samples were ultrasonically cleaned and dried with cold air (the cleaning medium and ultrasonication time were alcohol and 15 min, respectively). Next, their mass was weighed with the analytical balance (ME204E, Shanghai Mettler Toledo instrument Co., Ltd., Shanghai, China).

Salt spray corrosion tests were conducted with a salt spray tester (SK-60C, Wuxi Shangkai Test Equipment Co., Ltd., Wuxi, China) according to the ASTM-B117 standard (China). The temperature of the salt spray corrosion box was 35 ± 2 °C, and the salt spray sedimentation rate was 1–2 mL/(80 cm^2^·h). The corrosion medium was 5 wt.% NaCl solution prepared from analytically pure NaCl and deionized water with a pH value of 6.5–7.2. Samples were placed in the box, and the working surface was 20° from the vertical direction. Before the salt spray test, the box and sample were kept warm at 35 ± 2 °C for 2 h.

The dimensions of the samples used for the salt spray corrosion test were 15 mm × 15 mm × 6 mm. The salt spray corrosion timings were 0 min, 10 min, 0.5 h, 1 h, 3 h and 5 h. After the salt spray corrosion test, the corrosion surface of the samples was dried with cold air.

#### 2.2.2. Weight Loss Test

After the corrosion test, deionized water was used to dissolve the soluble salt on the surface. Concentrated nitric acid with a density of 1.42 g/mL was used to remove the corrosion products on the surface in compliance with the national standard GB/T 16545-2015 (China). The samples were also ultrasonically cleaned and dried with cold air. The mass of each sample was weighed with the analytical balance.

The corrosion rate of the samples can be calculated using Equation (1) [20,21]:(1)$$V_{t}=\frac{m_{1}-m_{2}}{ST}$$
where *V*_t_ is the corrosion rate, mg/(cm^2^·h); m_1_ is the mass of sample before test, mg; m_2_ is the mass of sample after test, mg; *S* is the test surface area of on one side of specimen, cm^2^; and *T* is corrosion time, h.

The surface morphology and the cross section microstructure of salt spray corrosion samples were observed by a scanning electron microscope (SU5000). In addition, the microstructures of the samples before the corrosion test were observed by a metallurgical microscope.

#### 2.2.3. Electrochemical Characterization

Electrochemical tests were carried out using an electrochemical workstation (CorrTestTM, Wuhan Kesite Co., Ltd., Wuhan, China). The test adopted a three-electrode system, with a Pt electrode as the counter electrode, a saturated calomel electrode as the reference electrode and the aluminum samples as the working electrode, with a working area of 1 cm^2^. The electrolyte solution was 3.5 wt.% NaCl solution. The scanning range of the polarization curve test was −0.35–0.35 V, and the scanning speed was 5 mV/s. The frequency range of electrochemical impedance spectroscopy (EIS) was 10^5^–10^−2^ Hz, and the amplitude of the applied sinusoidal disturbance wave was 5 mV. 

Mott–Schottky measurements were performed adopting −0.5−0.5 V vs. OCP with a scan rate of 10 mV/s at room temperature. The test frequency was 1000 Hz.

## 3. Results and Discussions

### 3.1. Corrosion Rate in Salt Spray Environment

Figure 1 shows the corrosion rate curve of the original and shot peen-formed aluminum samples in salt spray environment. With the extension of corrosion test time, the corrosion rate decreased rapidly in the initial stage and then gradually tends to be constant. In the initial stage (stage I), the corrosion rate was fast, and after 3 h (stage II), the corrosion rate was slow and stabilized. For the original sample, when the salt spray corrosion time was 10 min, the corrosion weight loss rate was about 2.8 mg/(cm^2^·h). When the corrosion time was 3 h, the corrosion weight loss rate decreased to about 0.21963 mg/(cm^2^·h). For the sample subjected to shot peen forming, when the salt spray corrosion time was 10 min, the corrosion weight loss rate was about 3.52001 mg/(cm^2^·h). When the corrosion time was 3 h, the corrosion weight loss rate decreased to about 0.38223 mg/(cm^2^·h). According to the above analysis, the corrosion rate of the shot peen forming sample was slightly faster than that of the original sample.

Citing the shot peen forming sample as an example, Figure 2a,b show the effect of salt spray corrosion time on the polarization curve and self-corrosion current density curve of this sample, respectively. As seen from Figure 2b, the corrosion current density of the shot peen-formed samples decreased with the extension of the corrosion time. The change trend of corrosion current density was consistent with the weight loss rate of this sample in Figure 1.

### 3.2. Corrosion Surface Morphology in a Salt Spray Environment

Figure 3 shows the morphology of the shot peen forming sample in neutral salt spray corrosion environment. Before the salt spray corrosion, there were many craters formed by the impact of projectiles on the surface of shot peen-formed sample. By dimensional measurement, the diameter of the crater was about 125.9 μm. Meanwhile, it can also be seen in the local enlarged view that microcracks appeared at the ridges where the craters intersected each other, as shown in Figure 3a. H. Kvoaci et al. [22] discovered that the morphology of the alloy after shot peening was characterized by deformation region and microcrack. The position of microcracks developed into corrosion pits during salt spray corrosion, and the pit diameter was about 14.8 μm, as shown in Figure 3b,c. Upon extending the salt spraying corrosion time, the size of corrosion pits became higher, the number of them increased and then the corrosion holes formed, as shown in Figure 3d–f. The research of Zhang et al. [23] found that shot peening increased the roughness of the alloy surface, and microcracks appeared at the pit ridge, therefore reducing the corrosion resistance of the alloy. When the salt spray time was 5 h, the dimension of the corrosion pit was maximum (about 188.9 μm), as shown in Figure 3f. In the meantime, the corrosion holes were interconnected, and the corrosion was relatively more serious. On the basis of the above analysis, after salt spray corrosion, the diameter of the pits became larger, the depth increased, the number became more and even micro pit aggregation occurred in the condition of long-time salt spray corrosion due to the accumulation of corrosive media. The corrosion morphology of the samples subjected to shot peen forming was the result of the coordination reaction of the surface crack and plastic deformation. Shot peen forming increased the number of surface cracks, which was conducive to the formation of micro cracks and pits, and reduced the salt spray corrosion resistance of the alloy.

Figure 4 shows the comparison of element contents from inside and outside the crater of the shot peen forming samples before salt spray corrosion and after different times of salt spray corrosion. From the figure, the content of Fe in the crater was always higher than that outside the crater, while the content of Cl outside the crater was higher than that inside the crater. This suggests that through shot peen forming, Fe remained on the surface of the sample and distributed unevenly. Then, it contacted with Al, and galvanic corrosion occurred in salt spray environment [17]. This was due to the potential difference between Al and Fe (the standard electrode potentials of Al and Fe were −1.66 V and −0.44 V, respectively). Fabijanic et al. [18] studied that nanocrystalline magnesium shot peened with ZrO, steel and Al_2_O_3_ as shot pellets. The results showed that there were residues including Zr, Fe, Al, Cr and O on the surface, and the self-corrosion current density increased by more than 10 times. The research team of Wen [24] found that the introduction of Fe concerning AA2024 aluminum alloy increased the corrosion rate by one order of magnitude, signifying that the corrosion resistance was worsened. From Figure 3, pitting pits mainly appeared at the pit ridge. This was essentially due to the following two reasons: First, the leakage of NaCl in the crater was caused by the inclination of the sample at a certain angle during salt spray corrosion, so the Cl content outside the crater was higher; second, through shot peen forming, microcracks appeared at the pit ridge, and Cl^−^ eroded the alloy matrix through microcracks. In addition, the content of Cl outside the crater increased upon extending the salt spraying corrosion time. Therefore, corrosion pits were more likely to appear at the ridge.

Figure 5 shows the morphology of the original sample in neutral salt spray corrosion environment. Before the salt spray corrosion, the surface of the original sample was mainly the trace left after machining, as shown in Figure 5a. The diameter and depth of pitting pits on the alloy surface became larger, the number of pitting pits became higher and even the phenomenon of pitting pit accumulation occurred upon extending the salt spraying corrosion time. When the corrosion time of the original sample was short (10 min and 0.5 h), the corrosion form of the sample was mainly pitting corrosion with shallow depth, as shown in Figure 5b,c and Figure 6(a1,a2). When the salt spray time was long (≥1 h), the depth and number of pitting pits on the surface of this sample increased, as shown in Figure 5d–f and Figure 6(b1,b2,c1,c2). When the salt spray corrosion times were 3 h and 5 h, microcracks appeared around the pitting pits and increased with the extension of corrosion time. This was due to the formation of corrosion products in the process of salt spray corrosion. Due to the large volume of corrosion products, stress was generated in the expansion process. With the combined action of stress and external corrosive medium, corrosion microcracks were formed around the pitting pits [25,26]. Wang et al. [27] studied the corrosion behavior of LY12 aluminum alloy in different atmosphere environment; they also found similar results.

### 3.3. Corrosion Layer Characteristics in a Salt Spray Environment

#### 3.3.1. Corrosion Layer Characteristics of the Shot Peen-Formed Sample in a Salt Spray Environment

Figure 7 shows the cross-sectional morphology of the sample subjected to shot peen forming without salt spray corrosion and after salt spray corrosion for 10 min, 0.5 h, 1 h, 3 h and 5 h. Figure 8 shows the elements plane distribution of the shot peen-formed sample with salt spray times of 0.5 h and 3 h. As can be seen from Figure 7a, EDS analysis of the sample showed that the precipitated phase of the alloy was Al_2_CuMg phase. The sample underwent intergranular corrosion after the salt spray corrosion for 10 min, as shown in Figure 7b. 

When the salt spray corrosion time was 0.5 h, the EDS analysis showed that there were local high concentrations of O and Cl in the aluminum sample matrix, which suggests that the degree of intergranular corrosion of the alloy was aggravated, and the NaCl depositing on the alloy surface started to erode the alloy matrix, as shown in Figure 8(b1,c1). Meanwhile, there was an accumulation of a small amount of O element on the alloy surface, which shows that corrosion products began to form on the alloy surface, as shown in Figure 8(b1). In addition, cracks appeared in the corrosion products at grain boundaries, as shown in Figure 7c.

When the salt spray corrosion time was 1 h, the cracks at grain boundary were more obvious and the corrosion products peeled off, as shown in Figure 7d. When the salt spray corrosion time was 3 h, the transgranular fracture of the sample can be seen through analyzing the local enlarged diagram of the sample. Simultaneously, the intergranular corrosion distribution was network, as shown in Figure 7e. Wang et al. [28] studied the galvanic corrosion of anodized 6061 aluminum alloy in an industrial-marine atmospheric environment; they also found that the intergranular corrosion distribution was network. The EDS analysis of the alloy demonstrated that there was aggregation of O and Cl elements, which indicated that the corrosion products were reformed on the surface after spalling. In addition, NaCl further deepened the corrosion to the alloy matrix, as shown in Figure 8(b2,c2). When the salt spray corrosion time was 5 h, the corrosion products formed on the alloy surface peeled off and cracks appeared at the peeling place, as shown in Figure 7f.

Because the radius of Cl^−^ was small, and it was an active anion and featured strong penetration and erosion ability [25], corrosion products were first formed on the alloy surface. Due to the intensification of intergranular corrosion, the corrosion products gradually gathered at the grain boundary and expanded. Due to the large volume of the corrosion products, the corrosion products peeled off when the salt spray corrosion time was 1 h [29]. The Cl^−^ depositing on the stripped surface reacted with Al_2_O_3_ on the alloy surface again [30], which destroyed the oxide film on the alloy surface and caused intergranular corrosion repeatedly. Cl^−^ gradually replaced OH^−^ in the corrosion product and reacted with Al(OH)_3_ insoluble in water to form AlCl_3_, a water-soluble corrosion product. The reaction process is shown in the following Equations [31]:Al(OH)_3_ + Cl^−^ → Al(OH)_2_Cl + OH^−^(2)
Al(OH)_2_Cl + Cl^−^ → Al(OH)Cl_2_ + OH^−^(3)
Al(OH)Cl_2_ + Cl^−^ → AlCl_3_ + OH^−^(4)

On the one hand, corrosion products reformed at grain boundaries, causing expansion. Due to different angles of intergranular corrosion between multiple grains, as shown in Figure 7e, the intergranular corrosion generated an expansion force, which intensified upon the increasing of corrosion degree [32], resulting in transgranular corrosion. On the other hand, the formed corrosion products promoted the formation of the expansion force due to their large volume. With the combined action of these two factors, the corrosion products peeled off along and through the grain boundary at the same time, as shown in Figure 7f. Corrosion products formed → expanded → peeled off → formed, and such a process appeared repeatedly.

#### 3.3.2. Corrosion Layer Characteristics of the Original Sample in a Salt Spray Environment

Figure 9 shows the cross sectional morphology of the original sample without salt spray corrosion and after salt spray corrosion for 10 min, 0.5 h, 1 h, 3 h and 5 h. Figure 10 shows the elements plane distribution of the original sample with salt spray times of 1 h and 3 h. As can be seen from Figure 9a, EDS analysis of the sample showed that the precipitated phase of the alloy was the Al_2_Cu phase. The sample underwent intergranular corrosion after salt spray corrosion for 10 min, as shown in Figure 9b, which was slighter than that of the sample subjected to shot peen forming under this test condition. When the salt spray corrosion time was extended to 0.5 h, the salt spray corrosion degree was slightly aggravated, as shown in Figure 9c.

When the salt spray corrosion time was 1 h, the EDS analysis of this sample shows that another phase precipitated from the matrix was Al_2_CuMg phase. This sample represented typical pitting characteristics, as shown in the red box of Figure 9d [33]. The degree of intergranular corrosion was further exacerbated, and the intergranular corrosion distribution was network, as shown in Figure 9d. The analysis of alloy elements on the surface showed that there were high concentrations of O and Cl on the inner surface of aluminum alloy matrix, which suggests that the corrosion products were composed of Al, O and Cl, as shown in Figure 10(a1,b1,c1). Meanwhile, NaCl depositing on the alloy surface started to erode the alloy matrix. The reaction equations in regard to the formation of corrosion products were the Equations (2)–(4). After the surface of the samples was penetrated by Cl^−^, pitting corrosion was formed. Because the pitting pit was a closed environment, NaCl gathered in the pitting pit and penetrated the sample, resulting in more serious intergranular corrosion.

Figure 9e shows the cross sectional morphology of the original sample after salt spray corrosion for 3 h. As shown in Figure 9e, the element plane analysis of the alloy shows that O and Cl elements were enriched at the grain boundary, which indicates that corrosion products were formed at the grain boundary, as shown in Figure 10(b2,c2). Meanwhile, the intergranular corrosion distribution was lamellar. When the salt spray corrosion time was extended to 5 h, due to the large volume of corrosion products and expansion force, the corrosion products peeled off on the fresh surface and the corrosion products were reformed along the grain boundary after peeling, as shown in Figure 9f. Corrosion products underwent a process of forming → expanding → spalling → forming and appeared repeatedly during salt spray corrosion.

There were two mechanisms of exfoliation corrosion: One was that the alloy reacted with hydrogen at or near the grain boundary, that is, hydrogen embrittlement [34,35], resulting in a crack at the grain boundary. The other was that the corrosion products insoluble in water gathered at the grain boundary to produce expansion stress, resulting in spalling corrosion. The coordination role of these two mechanisms in the process of spalling corrosion was complex, and a unified consensus has not been reached [36].

According to the above analysis, for the shot peen forming sample, when the salt spray corrosion time was 10 min, the sample underwent intergranular corrosion and the cracks appeared in the corrosion products at grain boundaries at 0.5 h. When the salt spray corrosion time was 1 h, the corrosion products peeled off and the intergranular corrosion distribution was network at 3 h. When the corrosion time was 5 h, the corrosion products peeled off again and the peeling rate of corrosion products was fast. For the original sample, when the salt spray corrosion time was 10 min, the degree of the intergranular corrosion was slighter than that of the sample subjected to shot peen forming under this test condition. And at 3 h, the intergranular corrosion distribution was lamellar. The peeling of corrosion products occurred at 5 h. Consequently, the peeling rate of corrosion products was slow. As a result, shot peen forming reduced the corrosion resistance of the aluminum alloy. 

Figure 11 shows the microstructure of samples before and after shot peen forming. From the figure, through shot peen forming, grain refinement occurred. Consequently, for the shot peen-formed sample, the intergranular corrosion distribution was network.

### 3.4. Effect of Shot Peen Forming on Electrochemical Properties of Alloys

#### 3.4.1. Electrochemical Impedance Spectroscopy Results

Figure 12 and Figure 13 are the EIS spectra of the shot peen forming and original sample after salt spray corrosion for different times, respectively. As can be seen from Figure 12a and Figure 13a, the Nyquist plots of the two samples were a semicircle to describe the arc resistance in the high-frequency region and an approximate straight line in the low-frequency region. This demonstrates that the corrosion process changed from charge transfer controlling to diffusion controlling [37,38]. As can be seen from Figure 12c and Figure 13c, the maximum phase angle of these two samples did not exceed 90°, which shows that there was a deviation between the electrochemical interface and the ideal capacitance [39].

Before the salt spray corrosion, the impedance values of the original sample were slightly higher than that of the shot peen-formed sample. During salt spray corrosion, the change law of impedance values of the two samples was similar. That was to say, upon extending the salt spraying corrosion time, the impedance values first decreased and then increased. For these two samples, when the salt spray time was 0 min, the impedance value was lowest, as shown in Figure 12b and Figure 13b.

In line with the electrochemical test results of the long salt spray corrosion of the two alloys at different time, the change trend in the early stage of corrosion was similar, but there was a great difference in the later stage of corrosion. The impedance value of the original sample increased significantly in the later stage of salt spray corrosion. Although the impedance value of the shot peen forming sample increased slightly in the later stage of corrosion, it was significantly lower than that of the original sample, as shown in Figure 12b and Figure 13b. With regard to the Nyquist plot, the capacitive resistance arc radius of the shot peen-formed sample was smaller than that of the original sample in the later stage of salt spray corrosion, as shown in Figure 12a and Figure 13a. As can be seen from Figure 3a, many microcracks appeared at the pit ridge on the surface of aluminum sample subjected to shot peen forming. In the later stage of salt spray corrosion, the microcrack at the pit ridge of the sample subjected to shot peen forming was the main path of salt solution eroding the matrix, which was also an important reason for more serious intergranular corrosion, thicker corrosion layer and lower impedance of the shot peen forming sample in the later stage of corrosion although the corrosion products played a certain protective role in the matrix. Therefore, shot peen forming reduced the corrosion resistance of the aluminum alloy.

#### 3.4.2. Mott–Schottky Results

The passive film on surface of metal materials exhibits semiconductor characteristics and plays an important role in corrosion resistance. The Mott–Schottky curve is an important method to study the semiconductor properties of passive film [40]. Researchers [41,42] have studied the corrosion behavior study of stainless steel using the Mott–Schottky curve. They found that the passive film indicated n-type semiconductor properties with high enough electrode potential. Other series capacitors, such as Helmholtz capacitance and surface state capacitance, can be ignored.

The measured capacitance corresponded to that of the space charge layer and was shown as the depletion layer. The relationship between *C*_SC_ and *E* can be expressed as the Mott–Schottky plots. The measurement capacitances of semiconductors were given by the following Equation [42]:(5)$$C_{\mathrm{SC}}^{-2}=\frac{2}{e\cdot N_{A}\cdot \varepsilon \cdot \varepsilon_{0}}\left(E-E_{\mathrm{fb}}-\frac{k\cdot t}{e}\right)$$
where *e* is the electron charge (1.602 × 10^−19^ C), *N*_A_ is the doping density of the passive film (cm^−3^), *ε* is assumed to be 10 [43], *ε*_0_ is the vacuum dielectric constant (8.854 × 10^−14^ F·cm^−1^), *E* is the applied potential (V), *E*_fb_ is the flat band potential (V), *k* is the Boltzmann constant (1.38 × 10^23^ J·K^−1^) and *t* is the test temperature (K). When tested at room temperature, $\frac{k\cdot t}{e}$ is about 25 mV, which is negligible.

Figure 14 shows the change of $C_{\mathrm{SC}}^{-2}$ and potential curves of aluminium alloy passive film at a potential of −0.5–0.5 V (vs. OCP) after the salt spray corrosion for 1 h and 5 h. With the range of −1.17–−1.0 V, $C_{\mathrm{SC}}^{-2}$ decreased with the potential, representing p-type semiconductor properties, as shown in Figure 14a. The density of the receptor (*N*_A_) was also calculated to further understand the semiconductive properties of the passive films. Through shot peen forming, the *N*_A_ value of the alloy increased after the salt spray corrosion for 1 h and 5 h, which suggests that the conductivity of the passive film was improved and the corrosion resistance was worsened, as shown in Figure 14b.

## 4. Conclusions

In this paper, the corrosion behavior of the shot peen-formed aluminum samples was studied in terms of the corrosion sensitivity, corrosion defects and corrosion layer. Based on the results in this work, the conclusions can be drawn as follows:The salt spray corrosion resistance of aluminum sample was worsened through shot peen forming, especially at the initial stage. When the salt spray corrosion time was 10 min, the corrosion rates of the sample subjected to shot peen forming and original sample were about 3.52001 mg/(cm^2^·h) and 2.8 mg/(cm^2^·h), respectively. Upon extending the salt spraying time, the corrosion rate of these samples decreased, but the corrosion rate of the shot peen forming sample was still faster than that of the original sample. Through shot peen forming, the *N*_A_ value of the alloy increased after the salt spray corrosion for 1 h and 5 h.Many corrosion pits were formed on the surface of the aluminum sample after salt spray corrosion. Compared with the original sample, for the shot peen-formed sample at the initial stage of salt spray corrosion, a large number of corrosion pits were mainly distributed at the ridge of craters. When the salt spray corrosion time was 1 h, the corrosion pit penetrated in some regions. When the salt spray time reached 5 h, the diameter and depth of the corrosion pit increased. This was due to the residue of elements in the crater and outside the crater, which intensified the corrosion of the alloy.Shot peen forming promoted the occurrence of intergranular corrosion and the spalling of corrosion products on the alloy surface. For the shot peen-formed sample, when the salt spray corrosion time was 10 min, the sample underwent intergranular corrosion and the cracks appeared in the corrosion products at grain boundaries at 0.5 h. When the salt spray corrosion time was 1 h, the corrosion products peeled off. The intergranular corrosion distribution was network at 3 h. When the corrosion time was 5 h, the corrosion products peeled off twice and the peeling rate of corrosion products was fast. For the original sample, when the salt spray corrosion time was 10 min, the degree of the intergranular corrosion was slighter than that of the sample subjected to shot peen forming. When the salt spray corrosion time was extended to 3 h, the intergranular corrosion distribution was lamellar. When the salt spray corrosion time was 5 h, the spalling of corrosion products occurred and the spalling rate of corrosion products was slow.

## Figures and Tables

**Figure 1 materials-15-08583-f001:**
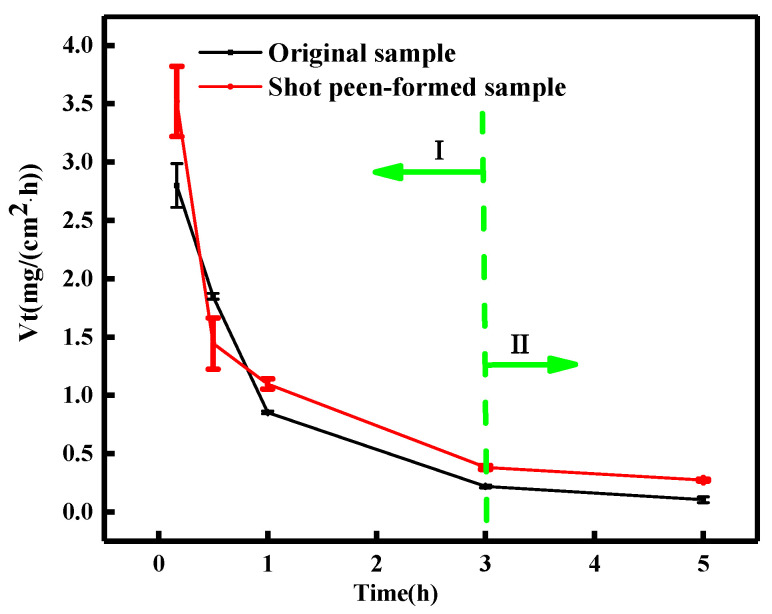
Comparison curve of the corrosion rate of the original and shot peen-formed sample. I is rapid corrosion stage. II is slow corrosion stage.

**Figure 2 materials-15-08583-f002:**
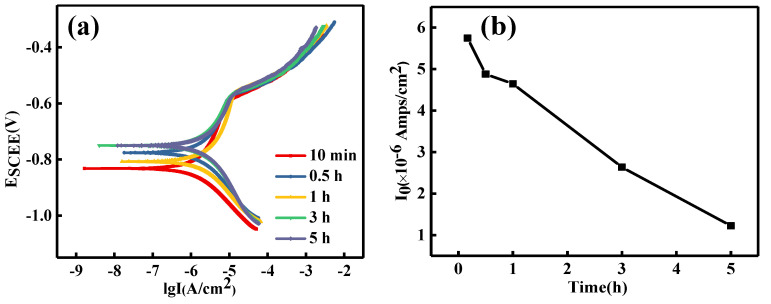
Effect of salt spray corrosion time on (**a**) the polarization curve and (**b**) the curve of self-corrosion current density of the shot peen-formed sample.

**Figure 3 materials-15-08583-f003:**
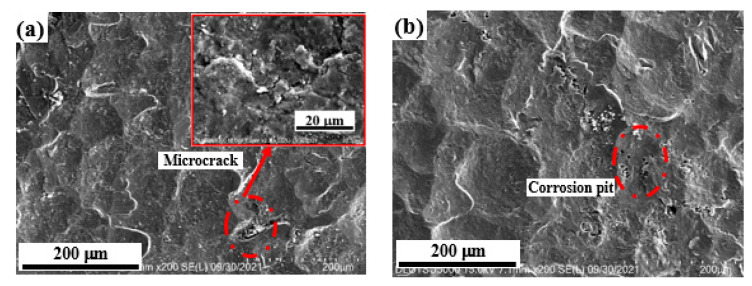
Corrosion surface morphology of the shot peen-formed samples in a neutral salt spray environment with salt spray times of: (**a**) 0 min; (**b**) 10 min; (**c**) 0.5 h; (**d**) 1 h; (**e**) 3 h; (**f**) 5 h.

**Figure 4 materials-15-08583-f004:**
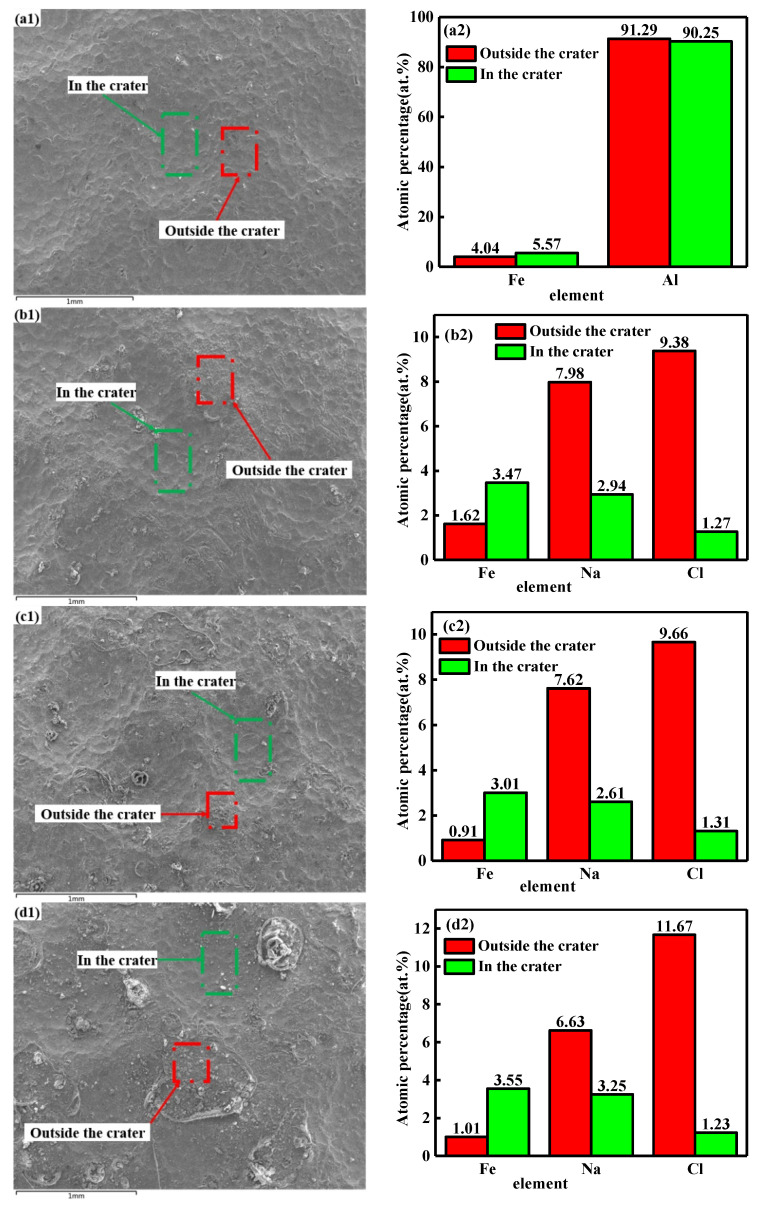
Comparison of element contents inside and outside the crater of the shot peen-formed samples in neutral salt spray environment with salt spray times of: (**a1**,**a2**) 0 min; (**b1**,**b2**) 0.5 h; (**c1**,**c2**) 1 h; (**d1**,**d2**) 3 h.

**Figure 5 materials-15-08583-f005:**
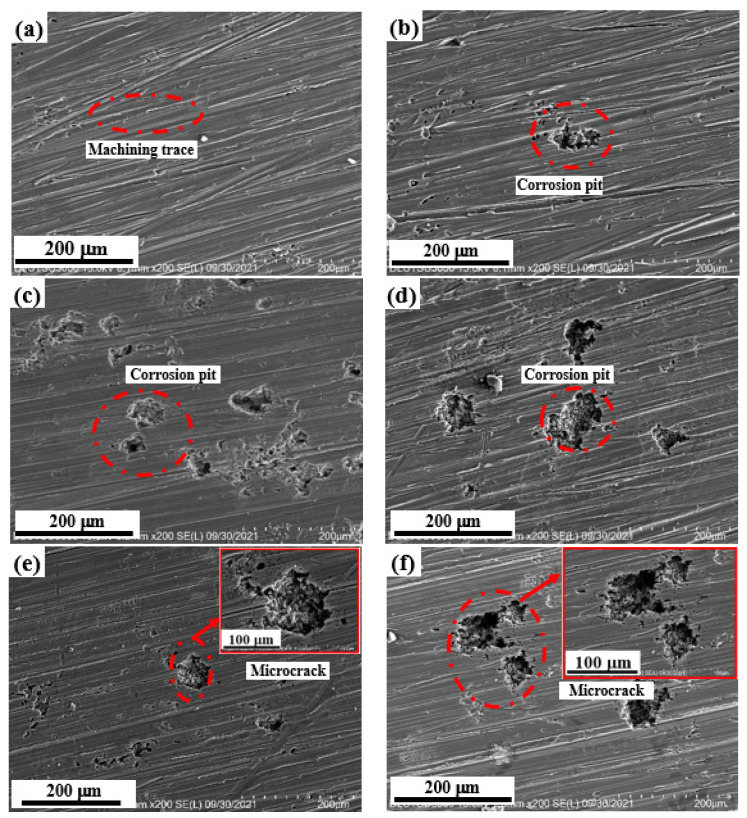
Corrosion surface morphology of the original samples in neutral salt spray environment after different times of: (**a**) 0 min; (**b**) 10 min; (**c**) 0.5 h; (**d**) 1 h; (**e**) 3 h; (**f**) 5 h.

**Figure 6 materials-15-08583-f006:**
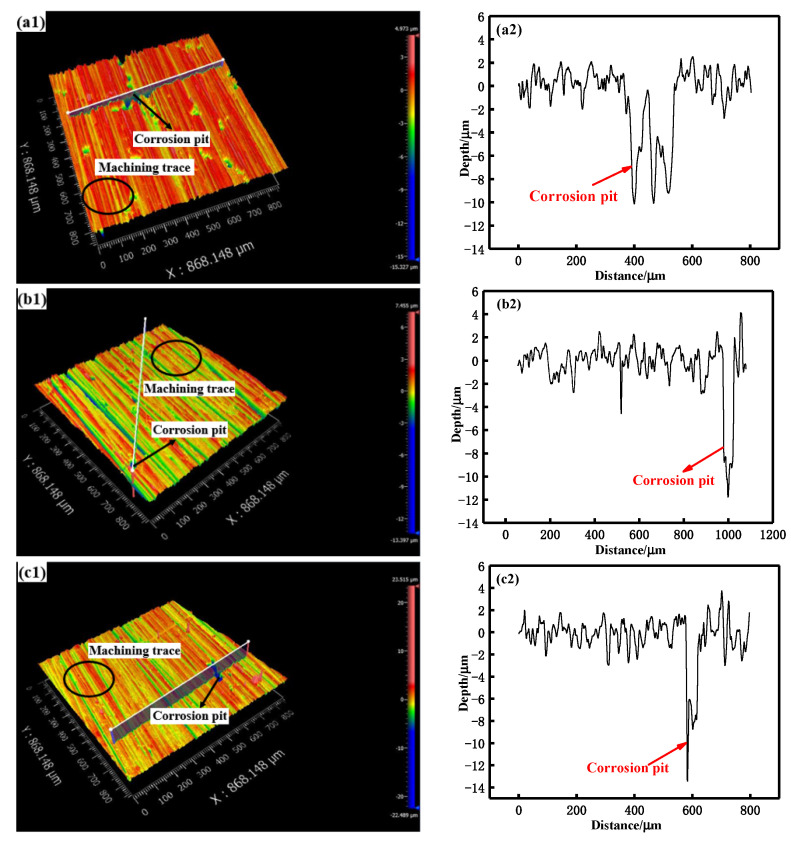
Corrosion pit depth of the original samples in neutral salt spray environment with salt spray times of: (**a1**,**a2**) 0.5 h; (**b1**,**b2**) 3 h; (**c1**,**c2**) 5 h.

**Figure 7 materials-15-08583-f007:**
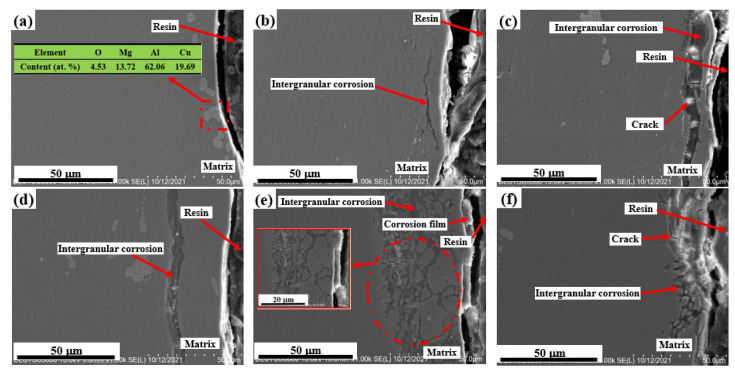
Cross section morphology of the shot peen-formed sample after salt spray corrosion: (**a**) 0 min; (**b**) 10 min; (**c**) 0.5 h; (**d**) 1 h; (**e**) 3 h; (**f**) 5 h.

**Figure 8 materials-15-08583-f008:**
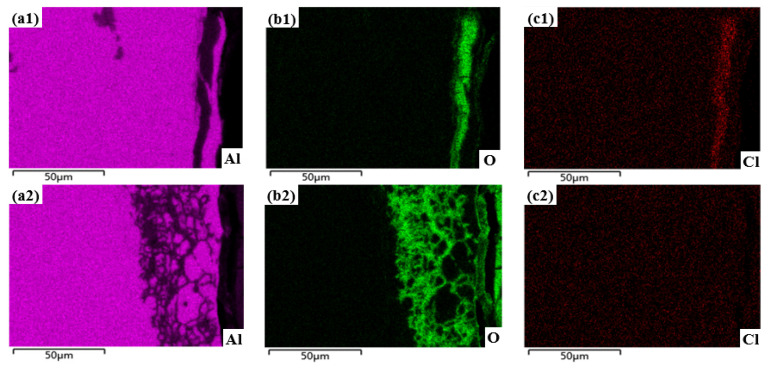
Elements plane distribution of the shot peen-formed sample with salt spray times of: (**a1**,**b1**,**c1**) 0.5 h; (**a2**,**b2**,**c2**) 3 h.

**Figure 9 materials-15-08583-f009:**
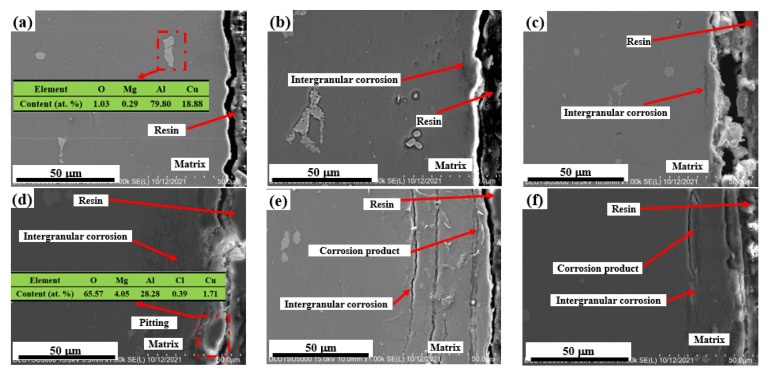
Cross section morphology of the original sample after salt spray corrosion: (**a**) 0 min; (**b**) 10 min; (**c**) 0.5 h; (**d**) 1 h; (**e**) 3 h; (**f**) 5 h.

**Figure 10 materials-15-08583-f010:**
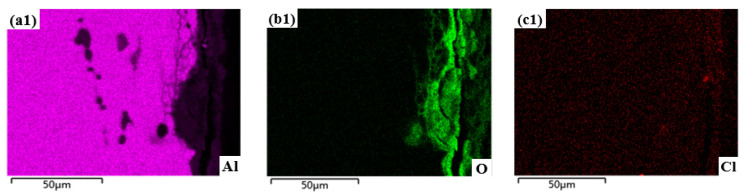
Elements plane distribution of the original sample with salt spray times of: (**a1**,**b1**,**c1**) 1 h; (**a2**,**b2**,**c2**) 3 h.

**Figure 11 materials-15-08583-f011:**
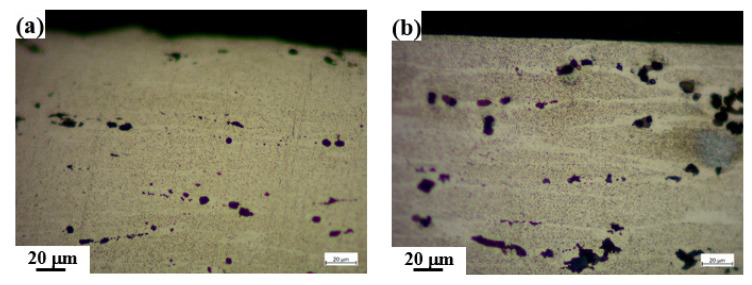
Microstructure of samples: (**a**) Shot peen-formed; (**b**) original.

**Figure 12 materials-15-08583-f012:**
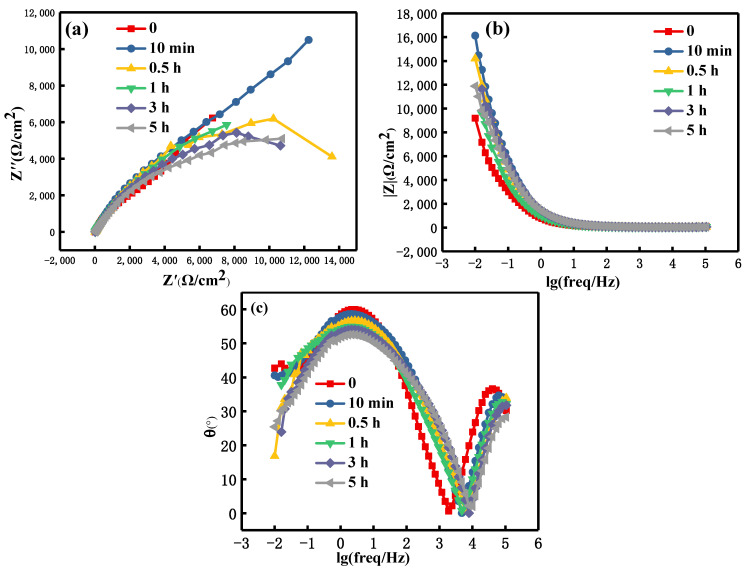
EIS spectra of the shot peen-formed sample after salt spray corrosion at different times in a neutral salt spray environment: (**a**) Nyquist plot; (**b**) Bode plot of modulus versus frequency; (**c**) Bode plot of phase angle versus frequency.

**Figure 13 materials-15-08583-f013:**
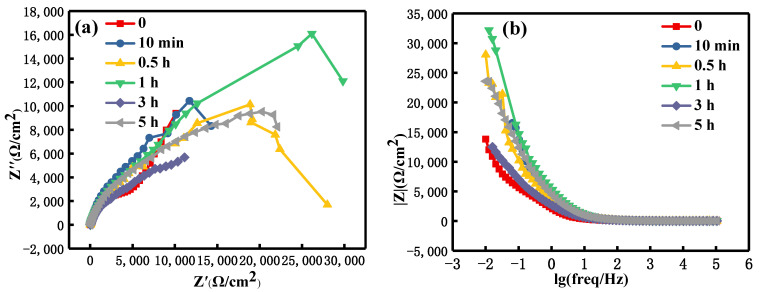
EIS spectra of the original sample after salt spray corrosion at different times in a neutral salt spray environment: (**a**) Nyquist plot; (**b**) Bode plot of modulus versus frequency; (**c**) Bode plot of phase angle versus frequency.

**Figure 14 materials-15-08583-f014:**
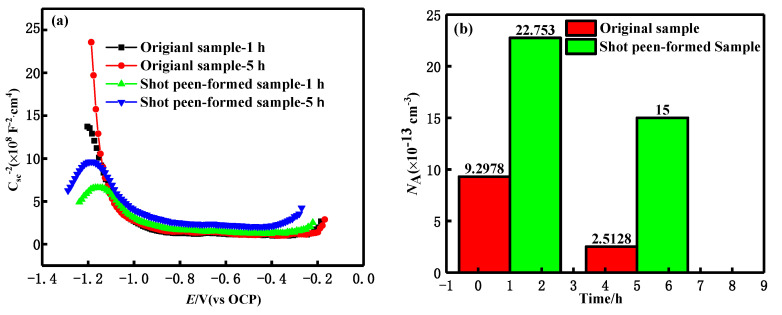
(**a**) Mott–Schottky plot and (**b**) *N*_A_ of these two samples in neutral salt spray environment with different salt spray times.

**Table 1 materials-15-08583-t001:** Chemical composition of the 2024 aluminum alloy (wt.%).

Element	Cu	Mg	Mn	Si	Fe	Zn	Ti	Cr	Al
Content	4.7	1.38	0.7	0.14	0.13	0.08	0.03	0.08	Bal.

**Table 2 materials-15-08583-t002:** Process parameters of shot peen forming and shot peen strengthening.

Process	Projectile Diameter/mm	Projectile Flow/(kg/min)	Jet Distance/mm	Shot-Peening Pressure/MPa	Coverage	Shot-Peening Strength/(mm A)
Shot peen forming	3	12	500	0.45−0.56	−	−
Shot peen strengthening	0.7	12	500	0.18	100%	0.15

## Data Availability

Not applicable.

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
