# Peer review of "Effect of Shot Peen Forming on Corrosion-Resistant of 2024 Aluminum Alloy in Salt Spray Environment"

_materials, 2022, doi:10.3390/ma15238583_

Round 1

Reviewer 1 Report

The authors have studied the effect of shot peen forming on corrosion resistance of Al2024 in salt spray environment. They have performed salt spray corrosion tests, weight loss tests, electrochemical characterization, surface morphology and microstructure analysis to support their conclusions. The study gives a complete insight into the corrosion resistance behavior of shot peened Al2024 material. 

The manuscript is written well. Language is clear and almost free from typographical and formatting errors. However, there are some minor corrections/clarifications which can be addressed such as,

1. Introduction section: good conductivity means thermal or electrical or both

2. Table 2: Kg should be kg

3. Section Introduction: It is mentioned Al2024 is a kind of deformed aluminum alloy. What does it mean?

Major concerns: 

1. Experiments are not repeated. Each experiment should be repeated 2-3 times and error bar should be included in the graphs. Without repetition, results has lower credibility.

2. Similar work is reported with different aluminum alloys also. Some works are already cited by the authors. The results are in similar line with the published work. Therefore, novelty of the work is not clear. 

The only visible novelty is that a different grade of aluminum alloy is used for this study and the results show that the findings of different aluminium alloys are valid for Al2024 also. Authors should clearly mention the novelty and why this study is important and should be considered for publication in this reputed journal. 

Reviewer 2 Report

The authors have nicely presented the research and its results, however, some modifications and improvements are required, below mentioned are my suggestions:

The introduction part of the manuscript seems a bit short. Would be good if the research problems is explained in details and possible solutions are discussed.

In line 86, ultrasonically cleaning is mentioned.. it would be good for the readers to mentioned cleaning medium and unltrasonication time. It may help readers in reproducing the results.

The result methodology is nicely presented, however, comparison to most recent literature is a bit weak

In Fig 14b, if the results are only for 7 seven days, then there is no need of mentioning 14 days.

It appears that either the font size or the space between the lines in Conclusion section is different from the rest of the ducments.

There is no literature citation from the year 2022 and only one from year 2021. I would suggest to add more recent and relevant literature for the result validation

Round 2

Reviewer 1 Report

Comments are addressed satisfactorily. 

Reviewer 2 Report

the authors have incorporated the changes in the manuscript, hence I suggest for the publication of the manuscript in its present form.